
# Entangled states of dipolar bosons generated in a triple-well potential

Arlei P. Tonel[1], Leandro H. Ymai[1], Karin Wittmann W.[2],
Angela Foerster[2] and Jon Links[3*]

**1** Universidade Federal do Pampa, Bagé, Brazil
**2** Instituto de Física da Universidade Federal do Rio Grande do Sul, Porto Alegre, Brazil
**3** School of Mathematics and Physics, The University of Queensland, Brisbane, Australia.

* jrl@maths.uq.edu.au

## Abstract

We study the generation of entangled states using a device constructed from dipolar bosons confined to a triple-well potential. Dipolar bosons possess controllable, long-range interactions. This property permits specific choices to be made for the coupling parameters, such that the system is integrable. Integrability assists in the analysis of the system via an effective Hamiltonian constructed through a conserved operator. Through computations of fidelity we establish that this approach, to study the time-evolution of the entanglement for a class of non-entangled initial states, yields accurate approximations given by analytic formulae.

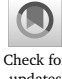
# 1 Introduction

Entanglement is a fundamental quantum resource, one which underpins many proposals for the implementation of quantum technology. Ultracold quantum gases have been viewed, for some time, as one of the most promising avenues for the physical production and manipulation of entangled states [1]. Experimental efforts towards achieving macroscopic entanglement continue to drive significant research activity, e.g. [2–7]. For ultracold quantum gases confined to triple-well potentials, many opportunities exist for the exploration of intriguing phenomena such as transistor-like behaviours [8–10], coherent population transfer [11, 12], fragmentation [13, 14], and quantum chaos [15]. Moreover, the use of dipolar atoms [16], with controllable long-range interactions, provides a platform for a rich array of physics including macroscopic cat states [17], stable quantum droplets [18], supersolid states [19], and prospects for interferometry [20].

In our recent work [21], we examined a special case of the general model of [20]. This restricted model belongs to a class of integrable tunneling models [22]. We identified within the integrable model a resonant tunneling regime, characterised by near-perfect harmonic oscillations with amplitude and frequency given by simple formulae. Through an appropriate breaking of the integrability, it was demonstrated how the amplitude and frequency could be varied in a predictable manner. This provided a design for a switching device, a fundamental component for the assembly of atomtronic circuitry (e.g. see [23]). Physical realisation of the system is feasible using dipolar atoms such as $^{52}$Cr or $^{164}$Dy. In this setting the atoms are confined using a system of lasers. Manipulation of the system is achieved by displacing the focus of a laser.

Our main objective in the present work is to expand on the analysis conducted in [21] in two complementary directions. The first of these is to investigate and understand the behaviour of the device with respect to a variety of initial conditions. In [21], the dynamical evolution was only considered for an initial state with all particles in one well. Here, we extend the investigations to the case of initial states with bosons distributed over the wells. It is found that the harmonic character of oscillations is still present for appropriately chosen interaction strengths, and the dynamics can be succinctly described. Encouraged by that result, we then turn attention to an analysis of the entanglement generated within the device under time-evolution. That is, our aim in this work is to input non-entangled states and analyse the capacity of the device to produce entangled states of an ultracold quantum gas. Due to the integrability of the system, many results can be obtained through analytic formulae.

The paper is organised as follows. The integrable Hamiltonian is introduced in Sect. 2, and an analysis is given for the energy spectrum. In Sect. 3 we conduct computations for the quantum population dynamics, and identify the resonant tunneling regime. We then introduce an effective Hamiltonian, which leads to analytic expressions for the frequency and amplitude of coherent oscillations between the outer wells. We also undertake numerical calculations for the quantum fluctuations. Sect. 4 deals with entanglement dynamics for different initial states, and in Sect. 5 we obtain an analytic expression for the time-evolution of states in the resonant regime. Sect. 6 contains final remarks.

# 2 Integrable Hamiltonian

An integrable model for a dipolar bosons loaded in an aligned triple-well potential was recently studied in [21]. In particular, the breaking of integrability was used to control the tunneling between the two external wells, thus implementing an atomtronic switching device. In this work, we investigate this model in more detail. The integrable triple-well Hamiltonian is given

by

$$H = U\left(N_1 - N_2 + N_3\right)^2 + J_1\left(a_1^\dagger a_2 + a_2^\dagger a_1\right) + J_3\left(a_2^\dagger a_3 + a_3^\dagger a_2\right). \tag{1}$$

It describes interactions between dipolar bosons in a triple well potential (with strength $U$), and tunneling between neighbouring wells (with strength $J_1$ and $J_3$). See Appendix A. The derivation of the Hamiltonian from first principles, and experimental feasibility, is discussed is in Appendix B. Hereafter, we set $J_1 = J_3 = J/\sqrt{2}$, and work in units such that $\hbar = 1$.

The Hamiltonian commutes with the total number operator $N = N_1 + N_2 + N_3$. For each fixed value of $N$, the Hamiltonian acts on a Hilbert space with dimension $d = (N+1)(N+2)/2$. Beyond the total number of bosons, the operator

$$Q = \frac{J^2}{2}\left(N_1 + N_3 - a_1^\dagger a_3 - a_1 a_3^\dagger\right) \tag{2}$$

is also conserved. We remark that $Q$ does not commute with the Hamiltonian if periodic boundary conditions are imposed.

First, we illustrate the structure of the energy levels. We fix the parameters $J = 1$ and $N = 20$ and only consider $U > 0$. In Fig.1, we plot the ordered energies for four choices of the interaction parameter $U$. For $U = 0$, there are $2N + 1$ distinct energy levels and a high level of degeneracy, with a uniform gap $\Delta E = J$ between adjacent levels. For $U \neq 0$ the energies mix and it is clearly seen that the energy spectrum undergoes qualitative changes as $U$ is varied. In particular, we observe the emergence of new energy bands, for sufficiently large values of $U$, with the gap between bands occurring at a larger energy scale.

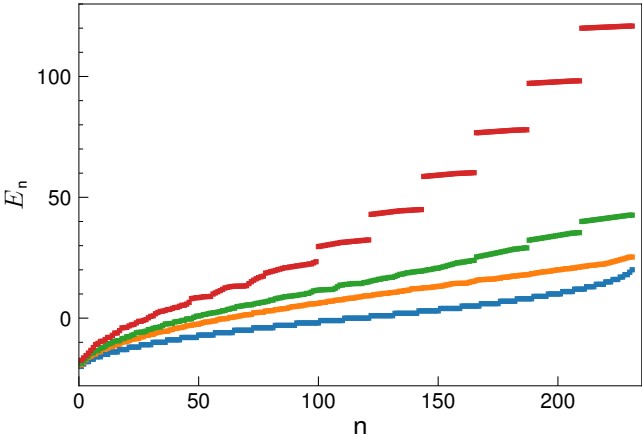

Figure 1: Ordered energy levels. Arranging the energy levels according to $E_n \leq E_m$ for n<m, the figure shows $E_n$ versus the level n. From top to bottom: $U = 0.3$ (red), $U = 0.1$ (green), $U = 0.05$ (orange) and $U = 0$ (blue), for $N = 20$.

A complementary presentation is provided in Fig. 2, where the energy level dependence is plotted against the dimensionless parameter $UN/J$.

At the intermediate interaction regime, $UN/J \sim 1$, a band structure begins to emerge (see the vertical line in Fig. 2). By increasing the dimensionless coupling parameter, the number of bands increases with non-uniform spacing between them. In the strong interaction regime $UN/J \gg N^2$, all energy levels tend to degenerate into bands. In the extreme limit $UN/J \to \infty$, the number of bands is $(N + 2)/2$ if $N$ is even and $(N + 1)/2$ if $N$ is odd.

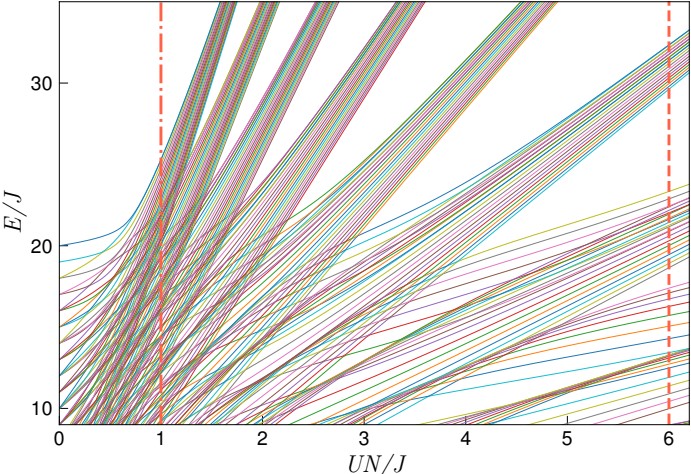

Figure 2: Energy level distribution. The dimensionless energies $E/J$ are plotted against dimensionless coupling parameter $UN/J$. At $UN/J = 0$ the energies are equidistant and degenerate. As $UN/J$ increases, the energies mix until reaching a nearly uniform distribution around $UN/J \simeq 1$ (the dot-dashed vertical line). Increasing $UN/J$ further leads to the re-emergence of energy bands.

## 3 Quantum population dynamics

The time evolution of any state is governed by

$$|\Psi(t)\rangle = \sum_{n=1}^{d} a_n \exp(-iE_n t)|\phi_n\rangle, \tag{3}$$

where $a_n = \langle\phi_n|\Psi_0\rangle$ for initial state $|\Psi_0\rangle$, and $\{|\phi_n\rangle\}$ is a set of normalised eigenvectors associated with the energy eigenvalues $\{E_n\}$. We will analyse the dynamical evolution for the following class of initial Fock states

$$|N-l-k, l, k\rangle = \frac{(a_1^\dagger)^{N-l-k}}{\sqrt{(N-l-k)!}} \frac{(a_2^\dagger)^l}{\sqrt{l!}} \frac{(a_3^\dagger)^k}{\sqrt{k!}}|0,0,0\rangle, \tag{4}$$

where $l = 0,1,2,\ldots,N$ and $0 \le k \le N-l$. These Fock states are the most general non-entangled, number-conserving, pure states. The expectation value of the population in each well is computed through

$$\langle N_i(t)\rangle = \langle\Psi(t)|N_i|\Psi(t)\rangle, \quad i = 1,2,3. \tag{5}$$

In [21], the dynamics of the population expectation values was studied for the initial state $|N,0,0\rangle$. There a resonant tunneling regime was identified for $UN/J \gg 1$, where near-coherent oscillations occurred between wells 1 and 3. By coherent, we mean oscillations in the expectation values which have the same waveform, the same frequency, and constant phase difference. This regime coincides with the emergence of energy bands as depicted in Fig. 2. Below we extend this analysis to the class of initial states (4). First, we find that each of the energy bands can be associated with labels $l$ and $N-l$. In particular, all states in each band have, approximately, the expectation value $\langle N_2\rangle \approx l$ or $\langle N_2\rangle \approx N-l$. Moreover, it is anticipated that there will still be oscillations between wells 1 and 3 for initial states (4). An example is given in Fig. 3.

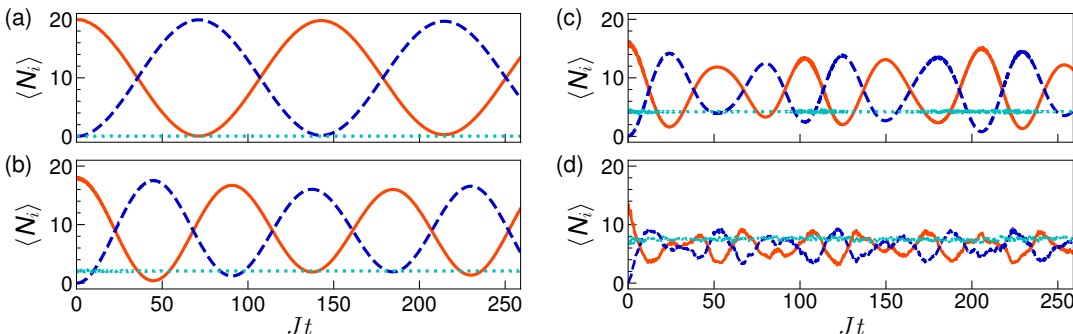

Figure 3: Expectation value dynamics. Dimensionless units are used. The panels show results for $\langle N_i \rangle$, $i = 1, 2, 3$, with $UN/J = 6$ and the choice of initial states: (a) $|20, 0, 0\rangle$; (b) $|18, 2, 0\rangle$; (c) $|16, 4, 0\rangle$ and (d) $|14, 6, 0\rangle$. $\langle N_1 \rangle$ is represented by the solid red line, $\langle N_2 \rangle$ by the dotted cyan line, and $\langle N_3 \rangle$ by the dashed blue line.

Fig. 3 shows two distinctive characteristics. The first is confirmation that $\langle N_2 \rangle$ is approximately constant for all choices of the initial states shown. The second, however, is that as the parameter $l$ in (4) is increased, there is a loss of coherence in the oscillations between wells 1 and 3. The reason for that loss of coherence can be appreciated from Fig. 4.

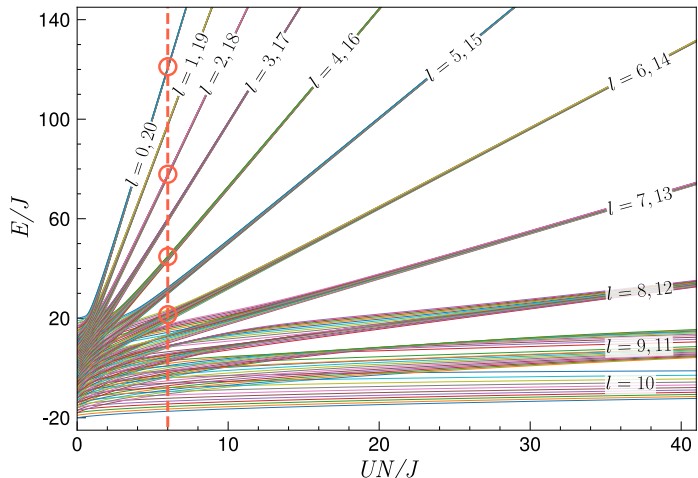

Figure 4: Energy level distribution. Dimensionless units are used. This figure is similar to Fig. 2, but displayed at a different scale. The labels shown on each band give the (approximate) possible values of $\langle N_2 \rangle$ for states within the band. The vertical line indicates $UN/J = 6$. The circles point out the four energy bands associated with the four initial states for the dynamics shown in Fig. 3.

To understand the above behaviour we define the sets $V_l = \{|N - k - l, l, k\rangle : k = 0, ..., N - l\}$. Then $V_l \cup V_{N-l}$ provides a basis for each band in the $UN/J \rightarrow \infty$ limit. When $UN/J$ is sufficiently large, but finite, these sets still provide accurate approximations for the bases. But as $l$ increases for $l \leq N/2$, or alternatively $l$ decreases for $l \geq N/2$, the threshold value of $UN/J$ which ensures well-separated bands increases. For $l = 0$ the band in Fig. 4 is clearly identifiable, whereas that for $l = 6$ is not. As we can see in Fig. 3, the dynamics for the initial state $|20, 0, 0\rangle$ leads to coherent oscillations, whereas the dynamics for $|14, 6, 0\rangle$ is not coherent.

The identification of approximate bases for the energy bands leads to a significant simplification in the computation of the dynamics, in the case when $UN/J$ is sufficiently large that the

bands are well separated. This is what we term the *resonant tunneling* regime. In this case the analysis is simplified through use of an effective Hamiltonian defined through the conserved operator $Q$. Using the technique of first-order and second-order transition processes [24, 25], we find that for an initial state of the form (4) the effective Hamiltonian is

$$H_{\text{eff}} = -\lambda_l Q, \tag{6}$$

where $Q$ is given by (2) and

$$\lambda_l = \frac{1}{4U}\left(\frac{l+1}{N-2l-1} - \frac{l}{N-2l+1}\right). \tag{7}$$

Using semiclassical analysis (see details in [21]), we obtain analytic expressions for the time evolution of the expectation value of populations in wells 1 and 3. Specifically

$$\langle N_1 \rangle = \frac{1}{2}\left(N - l + (N - l - 2k)\cos(\omega_l t)\right), \tag{8}$$

$$\langle N_3 \rangle = \frac{1}{2}\left(N - l - (N - l - 2k)\cos(\omega_l t)\right), \tag{9}$$

where $\omega_l = \lambda_l J^2$ is the frequency.

The formulae indicate that the initial population in well 3, i.e. $k$, does not affect the frequency of oscillation. However, it does impact on the amplitude. For $k = l = 0$ we recover the results discussed in [21] and maximum amplitude oscillations are attained. It is easily seen from (8,9) that equality of the expectation values $\langle N_1 \rangle$ and $\langle N_3 \rangle$ occurs when $t = (2n+1)T/4$, $n \in \mathbb{N}$, where $T = 2\pi/\omega_l$ is the period of oscillation. We confirm from numerical computations that this is also the case when the Hamiltonian (1) is used, rather than the effective Hamiltonian (6). See Fig. 5, where $N = 60$ is used to allow comparison with the results of [21]. For **(a)** – **(c)** the parameter $U = 0.17$ was chosen to lie in the resonant regime, as explained above (see Fig. 3). The value of $U = 0.7$ for the panels **(d)** – **(f)** was chosen such that the oscillations have, approximately, half the frequency of those in **(a)** – **(c)**.

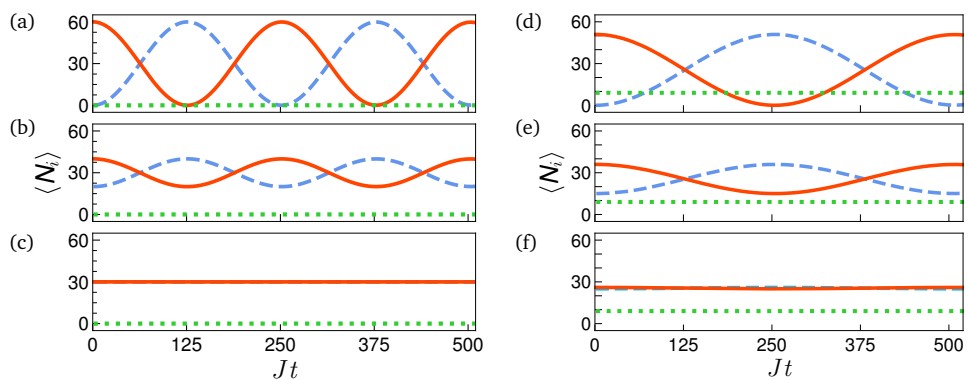

Figure 5: Expectation value dynamics. Dimensionless units are used. We set $J = 1$ and $N = 60$. The left panels are for $U = 0.17$ and the initial states : $|60, 0, 0\rangle$ [21], $|40, 0, 20\rangle$ and $|30, 0, 30\rangle$. The right panels are for $U = 0.7$ and the initial states: $|51, 9, 0\rangle$, $|36, 9, 15\rangle$ and $|26, 9, 25\rangle$. The solid red line is for $\langle N_1 \rangle$, the dotted green line is for $\langle N_2 \rangle$ and the dashed blue line is for $\langle N_3 \rangle$.

In order to gain insights into the quantum fluctuations of the coherent oscillations in the resonant regime, we now consider the variance of the expectation values. The variance $\sigma_j^2$ of the expectation value $\langle N_j \rangle$ is defined by

$$\sigma_j^2(t) = \langle N_j^2(t) \rangle - \langle N_j(t) \rangle^2. \tag{10}$$

Obviously, a semi-classical calculation leads to the result that the fluctuations are zero. Consequently, we compute values of the variance obtained through numerical diagonalisation of the Hamiltonian (1). Fig. 6 shows the normalised variance of $\langle N_1 \rangle$, $\sigma_1^2/(N-l)^2$, for the same initial states as in Fig. 5. For each $N$ (assumed even), the maximum amplitude for the variance occurs when wells 1 and 3 of the initial state are equally populated, while the minimum amplitude is obtained when one of the wells 1 or 3 is empty for the initial state. Observe that the period of the variance is $T/2$, where $T$ is the period of the expectation value oscillations in wells 1 and 3. The maximum amplitude occurs at the times $(2n+1)T/4$, $n \in \mathbb{N}$. These times will be seen to be significant in the subsequent discussion on entanglement.

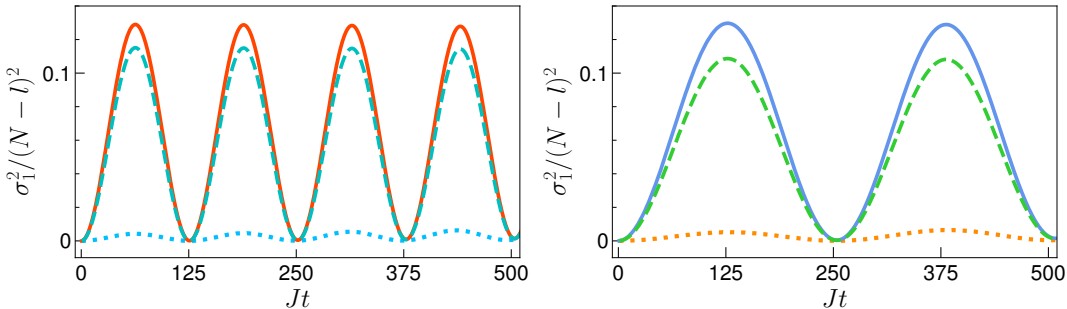

Figure 6: Time evolution of quantum fluctuations. Dimensionless units are used, and the fluctuations are normalised by $(N-l)^2$. We set $J = 1$ and $N = 60$. The left panel is for $U = 0.17$, $l = 0$ and initial states: $|60,0,0\rangle$ (dot blue line), $|40,0,20\rangle$ (dashed cyan line) and $|30,0,30\rangle$ (solid red line). The right panel is for $U = 0.7$, $l = 9$ and initial states: $|51,9,0\rangle$ (dotted orange line), $|36,9,15\rangle$ (dashed green line) and $|26,9,25\rangle$ (solid blue line).

## 4 Entanglement dynamics

For the following discussion we define the density matrix as

$$\rho(t) = |\Psi(t)\rangle\langle\Psi(t)|$$

and the reduced density matrices

$$\rho_1(t) = \text{tr}_2\,\text{tr}_3\,\rho(t), \qquad \rho_2(t) = \text{tr}_1\,\text{tr}_3\,\rho(t), \qquad \rho_3(t) = \text{tr}_1\,\text{tr}_2\,\rho(t),$$

where $\text{tr}_j$ denotes the partial trace over the space of states for well $j$. The von Neumann entropy, defined as [26]

$$S_j(\rho(t)) = -\text{tr}\left(\rho_j(t)\log\rho_j(t)\right) \tag{11}$$

provides a measure of the bipartite entanglement between the subsystem of well $j$ and the other two wells[1]. We also define the effective von Neumann entropy, which is calculated through the effective Hamiltonian (6). That is, for state evolution governed by

$$|\tilde{\Psi}(t)\rangle = \exp(-itH_{\text{eff}})|\Psi_0\rangle,$$

we define

$$\tilde{\rho}(t) = |\tilde{\Psi}(t)\rangle\langle\tilde{\Psi}(t)|,$$

---

[1]There is freedom to choose the base for the logarithm. Throughout, we use base 2.

and the reduced density matrices

$$\tilde{\rho}_1(t) = \text{tr}_2\,\text{tr}_3\,\tilde{\rho}(t), \qquad \tilde{\rho}_2(t) = \text{tr}_1\,\text{tr}_3\,\tilde{\rho}(t), \qquad \tilde{\rho}_3(t) = \text{tr}_1\,\text{tr}_2\,\tilde{\rho}(t),$$

where $\text{tr}_j$ denotes the partial trace over the space of states for well $j$. Then the effective von Neumann entropy is simply

$$S_j(\tilde{\rho}(t)) = -\text{tr}\,(\tilde{\rho}_j(t)\log\tilde{\rho}_j(t)). \tag{12}$$

Fig. 7 shows evolution of the entanglement between well 1 and the rest of the system, calculated from Eq. (11) for the Hamiltonian (1), and Eq. (12) for the effective Hamiltonian (6). The initial state is $|20,0,0\rangle$, and four values are shown for the parameter $UN/J$ from weak interaction (Fig. 7 (a)) into the resonant tunneling regime (Fig.7 (d)). In the latter case there is excellent agreement between $S_1(\rho(t))$ and $S_1(\tilde{\rho}(t))$. It is seen that the entanglement is a decreasing function of $UN/J$, indicating a tendency towards localisation. Note that the maximum entanglement that can be generated is $S^{\max} = \log d$ (dashed blue line), where $d = (N+1)(N+2)/2$. However, for the effective Hamiltonian the maximum entanglement that can be generated is $\tilde{S}^{\max} = \log(N-l+1))$ (dot-dashed blue line).

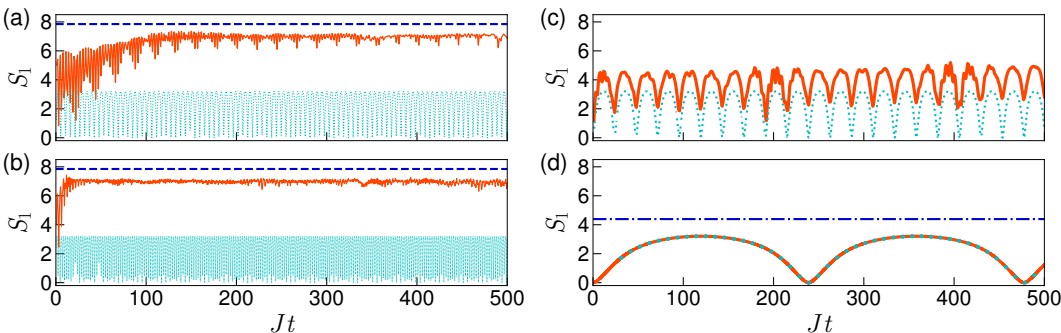

Figure 7: Entanglement dynamics. Dimensionless units are used. For all cases, $N = 20$ and $J = 1$, and the initial state is $|20,0,0\rangle$. (a): $UN/J = 0.02$, (b): $UN/J = 0.2$, (c): $UN/J = 2$, and (d): $UN/J = 20$. The orange lines depict $S_1(\rho(t))$ and the dotted cyan lines depict $S_1(\tilde{\rho}(t))$. The dashed blue lines (panels (a) and (b)) depict the maximum entanglement, while the dot-dashed line (panel (d)) represents the maximum entanglement that can be generated by the effective Hamiltonian (6).

In Fig. 8 the entanglement dynamics governed by the effective Hamiltonian is shown for six initial states in the resonant regime, the same as in Figs. 5 and 6. The maximum entanglement occurs in the vicinity of $t = (2n+1)T/4$ (These curves exhibit some irregular behaviours. It is difficult to precisely identify the times at which the maxima occur.). The times $t = (2n+1)T/4$ are also those for which the expectation values $\langle N_1\rangle$ and $\langle N_3\rangle$ are equal (Fig. 5), and their quantum fluctuations are maximal (Fig. 6).

## 5 Coherent state description

In this section we provide an explicit formula for states evolving under the effective Hamiltonian (6). These states belong to the class of $su(2)$ coherent states [27], and an expression for them can be compactly presented. Recall the Jordan-Schwinger representation of $su(2)$

$$\mathcal{J}_x = \frac{1}{2}(a_1^\dagger a_3 + a_1 a_3^\dagger), \qquad \mathcal{J}_y = -\frac{i}{2}(a_1^\dagger a_3 - a_1 a_3^\dagger), \qquad \mathcal{J}_z = \frac{1}{2}(N_1 - N_3),$$

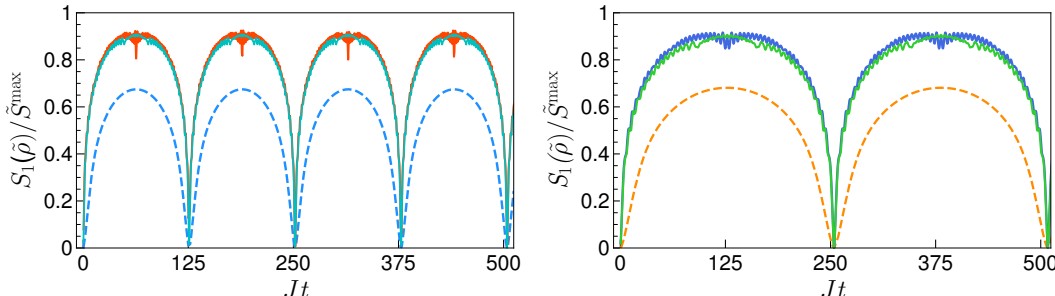

Figure 8: Entanglement dynamics under the effective Hamiltonian. Dimensionless units are used. We set $J = 1$ and $N = 60$. The left panel is for $U = 0.17$, $l = 0$ and initial states : $|60,0,0\rangle$ (dashed blue line), $|40,0,20\rangle$ (cyan line) and $|30,0,30\rangle$ (thicker red line). The right panel is for $U = 0.7$, $l = 9$ and initial states: $|51,9,0\rangle$ (dashed orange line), $|36,9,15\rangle$ (green line) and $|26,9,25\rangle$ (thicker blue line).

satisfying the commutation relations $[\mathcal{J}_a, \mathcal{J}_b] = i\epsilon_{abc}\mathcal{J}_c$. The time evolution operator $\tilde{U}(t) = \exp(-itH_{\text{eff}}) = \exp(i\omega_l t(N_1 + N_3)/2)\exp(-i\omega_l t\mathcal{J}_x)$ has the action (neglecting, without loss of generality, the overall phase $\exp(i\omega_l t(N_1 + N_3)/2)$)

$$\tilde{U}(t)a_1^\dagger \tilde{U}^\dagger(t) \propto \left(\cos(\omega_l t/2)a_1^\dagger - i\sin(\omega_l t/2)a_3^\dagger\right),$$
$$\tilde{U}(t)a_2^\dagger \tilde{U}^\dagger(t) \propto a_2^\dagger,$$
$$\tilde{U}(t)a_3^\dagger \tilde{U}^\dagger(t) \propto \left(-i\sin(\omega_l t/2)a_1^\dagger + \cos(\omega_l t/2)a_3^\dagger\right).$$

For initial state $|N-l-k,l,k\rangle$, the above expressions lead to the following form for the time-dependent state

$$|\tilde{\Psi}(t)\rangle = \tilde{U}(t)|N-l-k,l,k\rangle$$
$$\propto \sqrt{(N-l-k)!}\sqrt{k!}\sum_{j=0}^{k}\sum_{p=0}^{N-l-k}\frac{(-i)^{(j-p-k)}\sqrt{(j+p)!}}{p!(N-l-k-p)!}\frac{\sqrt{(N-l-p-j)!}}{j!(k-j)!}$$
$$\times (\sin(\omega_l t/2))^{(k+p-j)}(\cos(\omega_l t/2))^{(N+j-l-k-p)}|N-l-p-j,l,j+p\rangle \quad (13)$$

that can be rearranged to provide the wavefunction in a closed form (including the overall phase $\exp(i\omega_l t(N_1 + N_3)/2) = \exp(i\omega_l t(N-l)/2)$)

$$|\tilde{\Psi}(t)\rangle = \exp(it\omega_l(N-l)/2)\sum_{n=0}^{N-l}b_n(k,l,t)|N-l-n,l,n\rangle, \quad (14)$$

where $b_n(k,l,t)$ is given explicitly in Appendix C. Note that $\sum_{n=0}^{N-l}|b_n(k,l,t)|^2 = 1$ for all $t$.

While the formulae (13), (14) are not exact for the Hamiltonian (1), they provide an excellent approximation in the resonant tunneling regime. This is confirmed by calculations for the *fidelity*, $F$, defined by [28]

$$F = |\langle\Psi(t)|\tilde{\Psi}(t)\rangle|.$$

Illustrative examples are depicted in Fig. 9 below, where the parameter values are the same cases as those for Figs. 5, 6 and 8. The high values for the fidelity confirm the validity of the coherent state approximation. Note that the right panels in Fig. 9, where $l = 9$, are shown for a higher value of $U$ compared to $l = 0$ shown in the left panels. As mentioned earlier,

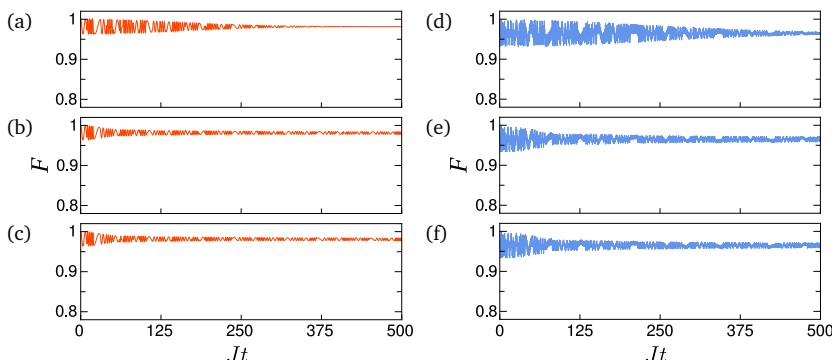

Figure 9: Fidelity as a function of time. Left panels: $U = 0.17$ and $l = 0$, for $N = 60$ and initial states **(a)**: $|60, 0, 0\rangle$, **(b)**: $|40, 0, 20\rangle$ and **(c)**: $|30, 0, 30\rangle$. Right panels: $U = 0.7$ and $l = 9$, for $N = 60$ and initial states **(d)**: $|51, 9, 0\rangle$, **(e)**: $|36, 9, 15\rangle$ and **(f)**: $|26, 9, 25\rangle$.

this higher value of $U$ is required in order to reach the resonant tunneling regime with well separated energy bands.

In principle, the coherent state expression (13) can be used to compute several important physical properties in the resonant tunneling regime for initial states (4). Below, the Fock state probabilities at $t = T/4$ are depicted. Each probability $|c_n|^2$ is associated with the basis vector $|60 - l - n, l, n\rangle$. These quantities are obtained through the dynamics of the Hamiltonian (1). The values closely match those given by the equivalent coherent state approximation through (14). The most probable states are: **(a)**: $|30, 0, 30\rangle$, **(b)**: $|58, 0, 2\rangle$ and $|2, 0, 58\rangle$, **(c)**: $|60, 0, 0\rangle$ and $|0, 0, 60\rangle$, **(d)**: $|26, 9, 25\rangle$ and $|25, 9, 26\rangle$, **(e)**: $|48, 9, 3\rangle$ and $|3, 9, 48\rangle$ and **(f)**: $|51, 9, 0\rangle$ and $|0, 9, 51\rangle$.

From these results one can identify that panels **(c)** and **(f)** show the states with the most uniform probability distribution. This helps to understand why the variance, and the entanglement entropy, increases with increasing number of particles $k$ in well 3 of the initial state. The decomposition of the state in terms of Fock states comprises an increasing number of components with increasing $k$. This correlates with the cases for $k = N/2$ ($N$ even), or $k = (N \pm 1)/2$ ($N$ odd) having the highest variance, as shown in Fig. 6, and the most entanglement, as shown in Fig. 8.

## 6 Conclusion

In this work we analysed the capacity for entanglement generation in an integrable, three-well atomtronic device. Our study was mostly undertaken in the resonant tunneling regime, where the particle exchange between wells 1 and 3 is very well described as coherent oscillation. We considered a class of unentangled initial states and studied the quantum evolution. We conducted calculations of population expectation values, quantum fluctuations, and entanglement. We found that the maximum entanglement occurred when the expectation values for populations in wells 1 and 3 are equal, where also their variances are maximal. We constructed an effective Hamiltonian, which is expressed in terms of a conserved operator for the integrable system. This allowed us to obtain an analytic formulae for the evolution of the states.

These results open a pathway towards new investigations, in two complementary directions. One of these is to understand the mechanisms for controlling the entanglement generation through the breaking of integrability, specifically through the inclusion of external

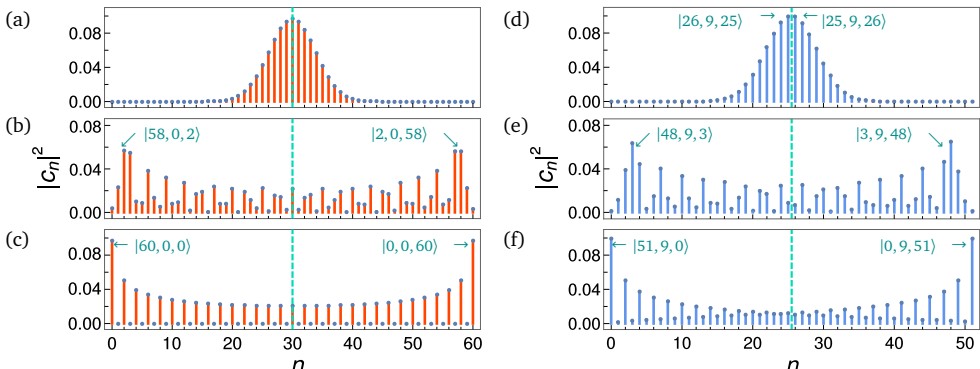

Figure 10: Fock state probabilities. Results shown for $t = T/4$ and $N = 60$. The index $n$ labels the Fock state $|N - l - n, l, n\rangle$, and $|c_n|^2$ denotes the probability associated to it. Left panels: $U = 0.17$ and $l = 0$ for initial states **(a)**: $|60, 0, 0\rangle$, **(b)**: $|40, 0, 20\rangle$ and **(c)**: $|30, 0, 30\rangle$. Right panels: $U = 0.7$ and $l = 9$ for initial states **(d)**: $|51, 9, 0\rangle$, **(e)**: $|36, 9, 15\rangle$ and **(f)**: $|26, 9, 25\rangle$. The vertical dashed lines mark the midpoints of the probability distributions. In the right column, where $l = 9$, the results display a slight asymmetry with respect to reflections about the midpoints.

fields as in [21]. Such an analysis of integrability breaking needs to encompass a study of the robustness of the system with respect to general deviations from integrable coupling. Some preliminary results are presented in [21], and this matter will be further investigated in a later, in depth, study.

The other avenue is to then examine the effects of external driving of the dynamics, which is feasible for this system using Floquet theory, e.g. see [29].

# Acknowledgements

J.L. acknowledges the traditional owners of the land on which The University of Queensland is situated, the Turrbal and Jagera people.

**Funding information** K.W.W. and A.F. were supported by CNPq (Conselho Nacional de Desenvolvimento Científico e Tecnológico), Brazil. A.F. acknowledges support from CAPES/UFRGS-PRINT. J.L. and A.F. were supported by the Australian Research Council through Discovery Project DP150101294.

# A  Integrable Hamiltonian

The extended triple-well Hamiltonian has the general structure [20]

$$\mathcal{H} = \frac{U_0}{2} \sum_{i=1}^{3} N_i(N_i - 1) + \sum_{i=1}^{3} \sum_{j=1; j \neq i}^{3} \frac{U_{ij}}{2} N_i N_j - J_1(a_1^\dagger a_2 + a_1 a_2^\dagger) - J_3(a_2^\dagger a_3 + a_2 a_3^\dagger). \quad (15)$$

Observing that $N^2 = (N_1^2 + N_2^2 + N_3^2) + 2N_1 N_2 + 2N_1 N_3 + 2N_2 N_3$ leads to

$$\mathcal{H} = \frac{U_0}{2}(N^2 - N) + (U_{12} - U_0)N_1 N_2 + (U_{13} - U_0)N_1 N_3 + (U_{23} - U_0)N_2 N_3$$
$$- J_1(a_1^\dagger a_2 + a_1 a_2^\dagger) - J_3(a_2^\dagger a_3 + a_2 a_3^\dagger). \quad (16)$$

Considering the particular case where $U_{13} = U_0$, and the symmetry configuration where $U_{12} = U_{23} = \alpha U_0$, the resulting Hamiltonian is integrable [21, 22] and can be written as

$$\mathcal{H}_0 = \frac{U_0}{2}(N^2 - N) + (\alpha - 1)U_0 N_2(N_1 + N_3) - J_1(a_1^\dagger a_2 + a_1 a_2^\dagger) - J_3(a_2^\dagger a_3 + a_2 a_3^\dagger),$$

where $\alpha$ is a constant parameter that depends only on the ratio $l/\sigma_x$, while $\sigma_x$ is the width of the Gaussian cloud along the $x$-direction. Now, setting $U = (\alpha - 1)U_0/4$, we demonstrate that the integrable Hamiltonian (1) is related to $\mathcal{H}$ through

$$
\begin{aligned}
H &= -\mathcal{H}_0 + (\alpha + 1)U_0 N^2/4 - U_0 N/2 \\
&= U(N_1 - N_2 + N_3)^2 + J_1(a_1^\dagger a_2 + a_2^\dagger a_1) + J_3(a_2^\dagger a_3 + a_3^\dagger a_2).
\end{aligned}
$$

## B Experimental feasibility

In order to discuss how to physically implement our proposal in a laboratory setting, and to provide numerical values of parameters for experimental setups in the cases of Chromium and Dysprosium, which produce the desired dipole-dipole coupling parameters, we follow the main lines of the discussion presented in [30]. For the general Hamiltonian that takes into account both contact and dipole-dipole interactions, we have

$$H = \int d^3\vec{r}\, \Psi^\dagger(\vec{r})(H_0)\Psi(\vec{r}) + \frac{1}{2}\int d^3\vec{r}\, d^3\vec{r'}\, \Psi^\dagger(\vec{r})\Psi^\dagger(\vec{r'})V(\vec{r} - \vec{r'})\Psi(\vec{r'})\Psi(\vec{r}), \qquad (17)$$

where

$$H_0 = -\frac{\hbar^2}{2m}\nabla^2 + V_{trap}(\vec{r})$$

for suitable trapping potential $V_{trap}$. The interaction potential is given by

$$V(\vec{r} - \vec{r'}) = V_{sr}(\vec{r} - \vec{r'}) + V_{dd}(\vec{r} - \vec{r'}),$$

where the short-range ($V_{sr}$) and the dipole-dipole interaction ($V_{dd}$) potentials have the form

$$V_{sr}(\vec{r} - \vec{r'}) = g\,\delta(\vec{r} - \vec{r'}),$$

$$V_{dd}(\vec{r} - \vec{r'}) = \frac{C_{dd}}{4\pi}\frac{(1 - 3\cos^2\theta)}{|\vec{r} - \vec{r'}|^3},$$

where $g = 4\pi\hbar^2 a/m$, $a$ is the $s$-wave scattering length, $C_{dd} = \mu_0\mu^2$, $\mu$ is the magnetic dipole moment, $\mu_0$ is the permeability of the vacuum and $\theta$ is the angle between the magnetic dipole moment (which is fixed in the $z$-direction) and the vector $(\vec{r} - \vec{r'})$. The scattering length $a(B) = a_{bg}(1 - \Delta B/(B - B_0))$ can be controlled by a magnetic field $B$, in the vicinity of a Feshbach resonance.

Consider the approximation

$$\Psi(\vec{r}) = \sum_{i=1}^{3} \phi_i(\vec{r})a_i,$$

where $\phi_i(\vec{r}) = w_0(\vec{r} - \vec{r_i})$ is the localized Wannier function in the center of well $i$ for a single particle. The localized Wannier function is the ground state of $H_0$ within a harmonic approximation.

Using the above approximations, the Hamiltonian (17) reduces to

$$H = \frac{U_0}{2} \sum_{i=1}^{3} N_i(N_i - 1) + U_{12}N_1N_2 + U_{23}N_2N_3 + U_{13}N_1N_3$$
$$- J_1(a_1^\dagger a_2 + a_2^\dagger a_1) - J_3(a_2^\dagger a_3 + a_3^\dagger a_2),$$

where

$$J_k = -\int d^3\vec{r}\, \phi_k^*(\vec{r}) H_0 \phi_2(\vec{r}),$$

$$U_{sr} = g \int d^3\vec{r}\, |\phi_1(\vec{r})|^4,$$

$$U_{dd} = \int d^3\vec{r}\, d^3\vec{r'}\, |\phi_1(\vec{r})|^2 |\phi_1(\vec{r'})|^2 V_{dd}(\vec{r} - \vec{r'}),$$

$$U_{ij} = \int d^3\vec{r}\, d^3\vec{r'}\, |\phi_i(\vec{r})|^2 |\phi_j(\vec{r'})|^2 V_{dd}(\vec{r} - \vec{r'}),$$

and $U_0 = U_{sr} + U_{dd}$. By symmetry $U_{12} = U_{23}$, and the Hamiltonian becomes

$$H = \frac{U_0}{2} \sum_{i=1}^{3} N_i(N_i - 1) + U_{12}N_2(N_1 + N_3) + U_{13}N_1N_3$$
$$- J_1(a_1^\dagger a_2 + a_2^\dagger a_1) - J_3(a_2^\dagger a_3 + a_3^\dagger a_2).$$

Using the conservation of $N$, we devolve into the same equation as Eq. (16).

A schematic representation of the experimental configuration is shown in Fig. 11. Three cigar-shaped Bose-Einstein condensates are trapped in a triple-well potential generated by three Gaussian beams, separated by a distance $l = 1.8\,\mu$m (one beam for each potential) along the $y$-axis. They are crossed by a transverse beam which provides $xz$-confinement. The (approximate) harmonic potential of each well $i = 1, 2, 3$ is symmetrically cylindrical and is locally given by

$$V_{trap}(x, y, z) = \frac{1}{2} m\omega_x^2 x^2 + \frac{1}{2} m\omega_r^2 ((y - y_i)^2 + z^2),$$

where $y_i = l, 0, -l$, $\omega_x$ is the frequency along the $x$-axis, and $\omega_r$ is the radial frequency in the $yz$-plane.

Table 1 lists experimental values, and resulting coupling parameters, for two different dipolar atoms, Chromium, $^{52}$Cr, and Dysprosium, $^{164}$Dy [21]. Some of the coupling parameters are indicated in Fig. 12.

## C  Coherent state approximation in closed form

Consider $\exp(-itH_{\text{eff}}) = VU$, where

$$V = \exp\left(\frac{i\omega_l t}{2}(N_1 + N_3)\right), \qquad\qquad \omega_l = \lambda_l J^2,$$

$$U = \exp(-i\omega_l t \mathcal{J}_x), \qquad\qquad \mathcal{J}_x = \frac{1}{2}(a_1^\dagger a_3 + a_3^\dagger a_1).$$

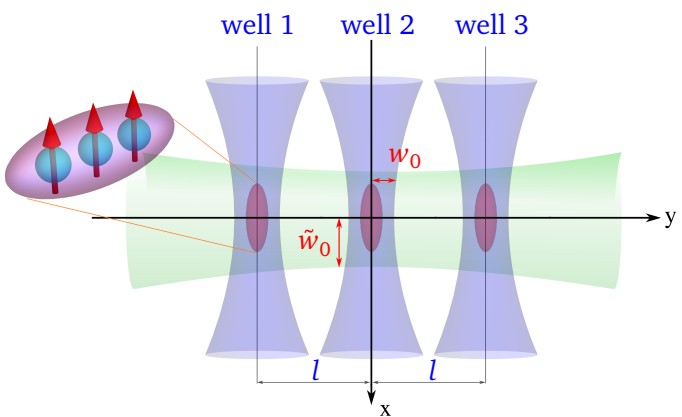

Figure 11: Schematic representation of the trap geometry. The three cigar-shapes (purple) represent the dipolar Bose-Einstein condensates, trapped in a triple-well potential formed by three parallel beams (blue), with waist $w_0$. The three beams are crossed by a transverse beam (green) with waist $\tilde{w}_0$. The wells are separated by a distance $l = 1.8\,\mu$m. The dipolar atoms are aligned to the $z$-axis by a magnetic field (red arrows).

Table 1: Experimental values and resulting parameters for $^{52}$Cr and $^{164}$Dy.

|  | Parameters | $^{52}$Cr | $^{164}$Dy |
|---|---|---|---|
| distance between wells | $l$ | 1.8 $\mu$m | 1.8 $\mu$m |
| parallel laser wavelength | $\lambda$ | 1.064 $\mu$m | 1.064 $\mu$m |
| parallel laser waist | $w_0$ | 1 $\mu$m | 1 $\mu$m |
| transverse laser wavelength | $\lambda$ | 1.064 $\mu$m | 1.064 $\mu$m |
| transverse laser waist | $\tilde{w}_0$ | 6 $\mu$m | 5 $\mu$m |
| $y-z$ radial trap frequency | $\omega_r/(2\pi)$ | 220 Hz | 67 Hz |
| $x$ trap frequency | $\omega_x/(2\pi)$ | 64 Hz | 23 Hz |
| $s$-wave scattering length | $a/a_B$ | 0.1 | 1 |
| dipolar scattering length | $a_{dd}/a_B$ | 16 | 131 |
| trap geometry parameter | $\alpha = U_{12}/U_0$ | 5.81 | 5.89 |
| one-site interaction | $U_0/(2\pi\hbar)$ | 0.019 Hz | 0.046 Hz |

Using the identity

$$\sum_{j=0}^{k}\sum_{p=0}^{N-l-k} f(p+j, p-j, p, j) = \sum_{n=0}^{N-l}\sum_{j\in S_n(k,l)} f(n, n-2j, n-j, j),$$

where

$$I_k = \{0, 1, 2, \cdots, k\},$$
$$S_n(k,l) = \{j \in I_k : j = n-p, \text{ where } p \in I_{N-l-k}\},$$

we find

$$U|N-l-k, l, k\rangle = \sum_{n=0}^{N-l} b_n(k,l,t)|N-l-n, l, n\rangle,$$

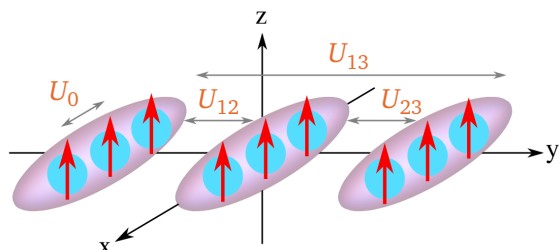

Figure 12: Schematic representation of the device. The cigar-shapes (purple), represent dipolar Bose-Einstein condensates, whose Gaussian distribution occurs predominantly in the $x$-direction. Dipole-dipole interactions are characterized by $U_{12} = U_{23} = \alpha U_0$, and $U_{13} = U_0$ is the integrability condition. Recall that $U = (\alpha - 1)U_0/4$ is the parameter that appears in the Hamiltonian (1).

with

$$b_n(k,l,t) = \sqrt{\frac{C_k^{N-l}}{C_n^{N-l}}} \sum_{j \in S_n(k,l)} (-i)^{k+n-2j} c_l^{N-l-k-n+2j} s_l^{k+n-2j} C_{n-j}^{N-l-k} C_j^k.$$

In the above expressions, the following shorthand notation is adopted for the binomial coefficients and trigonometric functions:

$$C_k^n = \binom{n}{k} = \frac{n!}{k!(n-k)!},$$
$$c_l = \cos(\omega_l t/2),$$
$$s_l = \sin(\omega_l t/2).$$

Note that, for all $t$,

$$\sum_{n=0}^{N-l} |b_n(k,l,t)|^2 = 1.$$

Thus, we have the normalised states

$$|\tilde{\Psi}(t)\rangle = \exp(it\omega_l(N-l)/2) \sum_{n=0}^{N-l} b_n(k,l,t)|N-l-n,l,n\rangle.$$

Using the above formula, we find

$$\langle \tilde{\Psi}(t)|N_1|\tilde{\Psi}(t)\rangle = \sum_{n=0}^{N-l}(N-l-n)|b_n(k,l,t)|^2 = \frac{1}{2}(N-l+(N-l-2k)\cos(\omega_l t)),$$
$$\langle \tilde{\Psi}(t)|N_2|\tilde{\Psi}(t)\rangle = \sum_{n=0}^{N-l}l|b_n(k,l,t)|^2 = l,$$
$$\langle \tilde{\Psi}(t)|N_3|\tilde{\Psi}(t)\rangle = \sum_{n=0}^{N-l}n|b_n(k,l,t)|^2 = \frac{1}{2}(N-l-(N-l-2k)\cos(\omega_l t)),$$

which agree with the semiclassical result presented in the main text.

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
