# Peer review of "Entangled states of dipolar bosons generated in a triple-well potential"

_SciPost Physics, doi:SciPost Phys. Core 2, 003 (2020)_

## Round 1 · Referee Report · Anonymous (Referee 1) · 2019-11-18

Report

In this paper the authors study the population and the entanglement dynamics for an integrable Hamiltonian describing a three-well Bose–Hubbard model obtained for a particular set of parameters. The time evolution sets in starting from non-entangled Fock states. What they predict is a coherent oscillation in the imbalance population between the first and the third well. This result is verified comparing it with an analytic calculation obtained in the large interaction regime, the so-called resonant tunneling regime. The paper is sound and well written, therefore I can recommend it for publication. However I would like the authors to clarify only few issues:

  • In Fig. 8 the maximum value of the entropy is 1. Is it the plot of S/S_max instead of S? (is the entanglement entropy normalized by its maximum value?)
  • Is there any reason why the time evolution of the entanglement entropy is lower for initial states with k=0, namely when the third well is empty at the initial time?
  • I would expect that the fidelity, defined as the overlap between the exact state and the one obtained by the coherent state approximation, approaches value 1 upon increasing U, namely going deeply in the coherent tunneling regime, while Fig. 9 does not show such expected behavior. Fig. 9 shows, instead, that the fidelity seems sensitive to the initial polulation of the second well. Could the authors comment on that?

---

## Round 1 · Referee Report · Anonymous (Referee 2) · 2019-12-23

Strengths

The paper studies the behaviour of the system with respect to a variety of initial conditions for the integrable Hamiltonian and it provides a clear analysis of the entanglement generated under time evolution.

Weaknesses

1) No particular effort is done to motivate the study: it is written that "the main goal is to expand on the analysis conducted in [18]", and explore a variety of initial conditions plus a study of dynamics. Despite it is understandable why this analysis may be desirable, the Authors do not elaborate enough on it in my opinion: what they expect from this study? why it is interesting? Since this analysis is restricted to states (4) where the number is defined, it is also clear that the analysis on these initial states is partial.

2) The Authors refer to their system as a "device": no effort is done to motivate the possible use in atomtronics, where, e.g., one would like to explore deviations from the integrable point. If from one side to call it "system" or "device" may be (to a certain extent) a matter of taste, at the same time it would be desirable to better motivate the use of the terminology "device", and may be to write "(e.g. see [20])" is not enough.

3) It would be important in my opinion to write the Hamiltonian of dipolar bosons in a trimer configuration, and then discuss under what specific fine tuning one gets the integrable Hamiltonian (1). If the Authors like to emphasize that "Physical realisation of the system is feasible using dipolar atoms such as 52 Cr or 164 Dy" they should describe how and if this tuning is realistically possible.

4) It would be important to see (or at least to mention or discuss) the possible effect of periodic boundary conditions a^\dag_1 a_3.

Report

I consider the paper interesting and without flows, on a deserving subject,
but I see that several points may be improved according the previous list.
I then suggest the Authors consider the previous points.

Requested changes

According the previous list of improvable points:

1) I suggest to improve the motivational point of the paper, and discuss what other initial conditions could have been interesting to explore and why they have been not considered (or perhaps left for future consideration?)

2) clarify the connection with atomtronics motivation - in that case please explain why the deviations from the integrable point may not spoil the analysis and the conclusions presented

3) present a derivation of (1) from the Hamiltonian of dipolar bosons in a three-well potential and discuss the fine tuning of parameters needed to obtain it

4) comment on the role of periodic boundary conditions

---

## Round 2 · Author Response

We thank the referees for their helpful comments, which have led to improvements in the presentation of the results.

---

## Round 2 · List of Changes

Response to Anonymous Report 1.

  1. The referee is correct that the entanglement has been normalised. We have modified the figure labels and caption text to reflect this.

  2. The following text has been included to answer the question raised by the referee.

From these results one can identify that panels (c) and (f) show the states with the most uniform probability distribution. This helps to understand why the variance, and the entanglement entropy, both increase with increasing number of particles $k$ in well 3 of the initial state. The decomposition of the state in terms of Fock states comprises an increasing number of components with increasing $k$. This correlates with the cases for $k=N/2$ ($N$ even), or $k=(N\pm 1)/2$ ($N$ odd) having the highest variance, as shown in Fig. 6, and the most entanglement, as shown in Fig. 8.

  1. As mentioned in the original submission, the resonant tunneling regime is dependent not only on $U$, but also on $l$. To quote

When $UN/J$ is sufficiently large, but finite, these sets still provide accurate approximations for the bases. But as $l$ increases for $l\leq N/2$, or alternatively $l$ decreases for $l\geq N/2$, the threshold value of $UN/J$ which ensures well-separated bands increases.

To further clarify this point we have added the following text in relation to Fig. 9

Note that the right panels in Fig. 9, where $l=9$, are shown for a higher value of $U$ compared to $l=0$ shown in the left panels. As mentioned earlier, this higher value of $U$ is required in order to reach the resonant tunneling regime with well separated energy bands.

Response to Anonymous Report 2.

  1. To clarify the issues raised by the referee, we have added the following text on page 2

That is, our aim in this work is to input non-entangled states and analyse the capacity of the device to produce entangled states of an ultracold quantum gas. Due to the integrability of the system, many results can be obtained through analytic formulae.

and on page 4

These Fock states are the most general non-entangled, number-conserving, pure states.

  1. The atomtronic realisation of the model is now described in Appendix B. We agree with the referee that the issue of the robustness of the system with respect to deviations from the integrable point is a very important point to investigate. Moreover, as was discussed in [21] the breaking of integrability is advantageous in the control of the tunneling oscillations. This will also be the case for the generation of entanglement. The text on page 11 has been modified to read

One of these is to understand the mechanisms for controlling the entanglement generation through the breaking of integrability, specifically through the inclusion of external fields as in [21]. Such an analysis of integrability breaking needs to encompass a study of the robustness of the system with respect to general deviations from integrable coupling. Some preliminary results are presented in [21], and this matter will be further investigated in a later, in depth, study.

  1. The information requested by the referee has been included in Appendices A and B.

  2. We do not consider the case of periodic boundary conditions for two reasons. The first is that periodic boundary conditions are not compatible with the experimental set up described in Appendix B. Secondly, the imposition of periodic boundary conditions breaks the integrability of the model. In particular, the operator $Q$ given by Eq. (2) does not commute with the Hamiltonian for periodic boundary conditions. The sentence

We remark that $Q$ does not commute with the Hamiltonian if periodic boundary conditions are imposed.

has been added on page 3.

---

## Editorial Decision

published